# “PhysIt”—A Diagnosis and Troubleshooting Tool for Physiotherapists in Training

**DOI:** 10.3390/diagnostics10020072

**Published:** 2020-01-28

**Authors:** Reuth Mirsky, Shay Hibah, Moshe Hadad, Ariel Gorenstein, Meir Kalech

**Affiliations:** 1Department of Software and Information Systems Engineering, Ben Gurion University, Negev 84105, Israel; shayhibah@gmail.com (S.H.); moshh@post.bgu.ac.il (M.H.); gorensar@post.bgu.ac.il (A.G.); kalech@bgu.ac.il (M.K.); 2Computer Science Department, University of Texas at Austin, Austin, TX 78712, USA

**Keywords:** model based diagnosis, applications, diagnosis, physiotherapy

## Abstract

Many physiotherapy treatments begin with a diagnosis process. The patient describes symptoms, upon which the physiotherapist decides which tests to perform until a final diagnosis is reached. The relationships between the anatomical components are too complex to keep in mind and the possible actions are abundant. A trainee physiotherapist with little experience naively applies multiple tests to reach the root cause of the symptoms, which is a highly inefficient process. This work proposes to assist students in this challenge by presenting three main contributions: (1) A compilation of the neuromuscular system as components of a system in a Model-Based Diagnosis problem; (2) The *PhysIt* is an AI-based tool that enables an interactive visualization and diagnosis to assist trainee physiotherapists; and (3) An empirical evaluation that comprehends performance analysis and a user study. The performance analysis is based on evaluation of simulated cases and common scenarios taken from anatomy exams. The user study evaluates the efficacy of the system to assist students in the beginning of the clinical studies. The results show that our system significantly decreases the number of candidate diagnoses, without discarding the correct diagnosis, and that students in their clinical studies find *PhysIt* helpful in the diagnosis process.

## 1. Introduction

When a patient contacts a physiotherapist (PT) regarding a problem in the peripheral nervous system or muscular system, the usual cues are either in terms of motion or sensory abilities. The patient can report some difficulty in performing a specific movement or a sensory problem such as numbness or tingling. A weakened motion is indicated by an observation on the muscles, while a defected sensation is indicated by an observation on the dermatomes. These reports are the symptoms of the patient. Based on the reported symptoms, the PT hypothesizes the possible reasons that could explain the patient’s complaints. Theses reasons are called diagnoses. To discriminate the root cause among the possible diagnoses, a troubleshooting process is executed in which the PT performs a series of tests that are meant to disambiguate between the correct diagnosis and the rest. This approach is usually time consuming and can be ineffective, especially in the case of trainee PTs with little experience. For example, some clinicians move back and forth between their original and revised hypotheses to come up with a final diagnosis [1].

This paper presents an AI-based tool—*PhysIt*—that enables an interactive visualization and diagnosis to assist trainee physiotherapists (The system can be viewed using the following link: http://www.ise.bgu.ac.il/PHYSIOTHERAPY/Homepage.aspx).

The first step to create such a tool is a compilation of the neuromuscular system as components of a system in a Model-Based Diagnosis problem. Our approach is based on modeling the physiotherapy diagnosis process as a model-based diagnosis (MBD) problem [2,3,4,5]. MBD relies on a model of the diagnosed system, which is utilized to simulate the expected behaviour of the system given the operational context (typically, the system inputs). The resulting simulated behaviour (typically, the system outputs) are compared to the observed behaviour of the system to detect discrepancies that indicate failures. The model is then used to pinpoint possible failing components within the system. In the physiotherapy domain, the observed system behaviour is the patient’s weakened motion or defected sensation. The system model is a model of the human body, such as the nervous system, the muscles, the dermatomes etc., as well as the connections between them. A diagnosis is the human body component(s) that does not function well. Modeling this problem as an MBD enables solving it by applying off-the-shelf MBD algorithms. This compilation is the first main contribution of this paper.

Second, the Physit system computes the diagnoses based on the observations and then it operates a troubleshooting algorithm to assist the PT to choose informative tests and finally identify the root cause of the patient’s complains. This is the second contribution of this paper and it can be decomposed into the different features of the Physit system:

The first feature of *PhysIt* is an interactive graphical model of anatomical entities. To this aim, we used expert knowledge to define the important entities that are required to clinically diagnose patients. In particular the tool focuses on nerve roots, nerves, muscles and dermatomes. Using this domain representation, we implemented an interactive inference to visually present the relationships between the entities.

The second feature of *PhysIt* is a framework to assist a trainee PT with the diagnosis process. This framework proposes the next recommended test to perform given the current state and observations given from the patient and previous tests. This framework utilizes the MBD algorithm described earlier.

The third feature of *PhysIt* is a troubleshooting process in which the root cause of the symptom is recognized. This is done by adapting an iterative probing process from the MBD literature [6], in which tests are iteratively proposed to the PT in order to eliminate redundant diagnoses.

There is a huge research on diagnosis in medicine. Most of the works propose frameworks and algorithms utilizing different diagnosis approaches, such as knowledge-based [7], data driven [8] and model-based [9]. Many of the works even run experiments on specific medical problems. As far as we know, no previous work presents a comprehensive tool for trainee PTs that includes visualisation, diagnosis and troubleshooting. Our work does not present new diagnosis or troubleshooting methods, but it utilizes previous model-based methods to present a tool that helps trainee PTs in the diagnosis process, by applying anatomical model visualization, diagnosis and troubleshooting.

The third contribution of this paper is a comprehensive performance analysis and a user study to evaluate *PhysIt*. The performance analysis was performed on both simulated cases and scenarios depicted by the domain experts, that are common cases in anatomy exams. We examined the diagnosis process in terms of accuracy, precision, waste costs and the AUC of the health state [10,11]. Our results show that the tool always finds the correct diagnosis and that the troubleshooting process can significantly decrease the number of candidate diagnoses, and thus facilitates trainee PTs. Our user study included simulations of a physiotherapy diagnosis process performed by physiotherapy students. The students were given different levels of access to our system, and were then requested to answer a questionnaire in order to evaluate the experience with the system. The study shows that our system was perceived as helpful in choosing the tests to perform and in improving the diagnosis process.

To summarize, this paper contains an exploratory research on the application domain of physiotherapy diagnosis. It consists of a new theoretical representation, based on MBD literature, to model a PT diagnosis process, and then utilizes this representation to propose an educational tool for PTs in the beginning of their clinical studies. This research proposes a thorough analysis of the proposed tool, both by testing its mathematical and theoretical capabilities, and by testing it empirically on use-cases used in physiotherapy exams and in a user study with PTs in training.

The flow of the paper is as follows: in the next section we detail the related work, then in Section 3 the architecture and interface of the tool will be presented. Section 4 describes technical details about the different parts of *PhysIt*: the model, the diagnosis algorithm and the troubleshooting process. In Section 5 the diagnosis and the troubleshooting processes will be evaluated and in Section 6 the user study will presented. Section 7 concludes this work.

## 2. Related Work

In Section 2.1 we present the main approaches for diagnosis and specifically model-based diagnosis methods in medicine. Then in Section 2.2 we depict troubleshooting approaches. Finally in Section 2.3 the contributions of our work are presented in the light of previous work.

### 2.1. Diagnosis

Diagnosis approaches are typically divided into three categories: data-driven, model-based, and knowledge-based. Data-driven approaches are model free. The online monitored data is used to differentiate a potential fault symptom from historically observed expected behaviour, e.g., via Principle Component Analysis [12]. Model-based approaches [13,14,15] typically use reasoning algorithms to detect and diagnose faults. The correct/incorrect behaviour of each component in the system is modeled as well as the connections between them, and the expected output is compared to the observed output. A discrepancy between them is exploited to infer the faulty components. Knowledge-based [16] approaches typically use experts to associate recognized behaviours with predefined known faults and diagnoses. A similar partition is proposed by Wagholikar et al. [17], which survey paradigms in medical diagnostic decision support, dividing most works into probabilistic models (Bayesian models, fuzzy set theory, etc.), data driven (SVM and ANN) and expert-based (rule-based, heuristic, decision analysis, etc.).

The decision on the best approach is obviously dependent on the domain knowledge. If we have enough data on past processes of the system then probably we would prefer to use data driven approaches, on the other hand if the system can be represented by rules, designed by experts, then a knowledge-based approach is preferred. Finally, if we can formally model the system, then a model-based approach will be appropriate. Next, we present relevant research and elaborate on MBD approaches within the context of medical systems.

In this paper we focus on a model-based approach, since we used expert physiotherapists which helped us to model the upper part of the human body which is innervated by the nerve roots C−3 to T−1. For a survey of knowledge-based approaches in medicine we refer the reader to [7]. There are additional surveys that address knowledge-based approaches in specific medical fields as breast cancer diagnosis [18] and medical expert systems for diabetes diagnosis [19]. Data driven approaches are very common in medicine, Patel et al. [8] and Tomar et al. [20] survey many of these approaches, specifically Kourou et al. survey machine learning approaches for cancer prognosis [21]. Data Mining techniques are used to label specific conditions such as Parkinson Disease [22] or Diabetes [23].

There are several approaches in MBD. All are relevant also to diagnosis in medicine [9,24]. They differ in the way the domain knowledge is represented. Obviously, in many cases the model is determined by the type of knowledge we have. Consistency-Based Diagnosis (CBD) assumes a model of the normal behaviour of the system [2,3]. Causality models describe a cause-effect relationships. There are two diagnosis approaches to deal with causality models, set-covering theory of diagnosis [25] and abductive diagnosis [26,27]. A third way to model a system is by a bayesian network [28], where the relations between the components are represented by conditional probability tables. Given evidence, an inferring process is run and produces a diagnosis with some probability. We survey each one of these approaches next.

General Diagnostic Engine (GDE) is an algorithm to solve the CBD problem [2]. This algorithm proceeds in two steps: (1) First, it finds conflicts in the system by using assumption-based truth maintenance system (ATMS) [29]. A conflict is a set of components, which when assumed healthy the system theory is inconsistent with the observation. (2) Then the GDE computes the hitting sets of the conflicts, where each hitting set is actually a diagnosis. Downing [30] proposes a system which extends GDE to deal with the physiological domain. For this, Downing extends the GDE to cope with (1) dynamic models by dividing the time to slices and solve the diagnosis problem for each slice, and with (2) continuous variables by representing the variables qualitatively. He gives some examples from the physiological domain such as diagnosing the stages of acidosis regulation. Also Gamper and Nejdl [31] cope with the temporal and continuous behaviour of medical domain. They propose to represent the temporal relationships between qualitative events in first-order logic and then, given observations, they run CBD algorithm to diagnose the system. They run experiments on a set of real hepatitis B data samples.

CASNET [32] is one of the pioneer causal models in medicine. It describes pathophysiological processes of disease in terms of cause and effect relationships. The relationships between the pathophysiological states are associated also with likelihood to direct the diagnosis. CASENT even links a therapy recommendation to the diagnostic conclusion. INKBLOT [33] is an automated system which utilizes neuroanatomical knowledge for diagnosis purposes. The model includes hierarchy anatomical model of the central nervous system where the cause effect relationships describe the connections between and damages and manifestations. Also Wainer et al. [34] describe a cause-effect model where the causes are disorders and effects are the manifestations. They extend the diagnostic reasoning, using Parsimonious Covering Theory (PCT) [35], to deal with temporal information and necessary and possible causal relationships between disorders and manifestations. They demonstrate their new algorithm on diagnosis of food-borne diseases.

The problem of diagnosis, often shown as a classic example of abductive reasoning, is highly relevant to the medical domain [36]. As shown in previous papers [27,37], abduction with a model of abnormal behaviour is much better way than consistency-based to deal with medical diagnosis. However, not always such knowledge is easy to obtain, since it requires experts to model not only the normal behaviour, but also how a component behaves in each one of its abnormal cases. Obviously, this knowledge helps to focus on more meaningful diagnoses, but it is difficult to obtain. Pukancová et al. [38] focus on a practical diagnostic problem from a medical domain, the diagnosis of diabetes mellitus. They formalize this problem, using information from clinical guidelines, in description logic in such a way that the expected diagnoses are abductively derived. The importance of taking into consideration temporal information in medicine has been previously recognized. Console and Torasso [39] discuss the types of temporal information which can be represented by causal networks, and they use a hybrid approach to combine abductive and temporal reasoning for the diagnosis process.

Bayesian networks (BN) is a probabilistic model using for diagnosis in various domains such as vehicles [40], electrical power systems [41] and network systems [42,43]. BN describes conditional probabilities between the components; given evidence (observations), an inference algorithm is used to compute the probability of each healthy component to propagate the evidence. A classical work in the medical domain is the Pathfinder, which is designed to diagnose lymphatic diseases using Bayesian belief networks. It begins with a set of initial histological features and suggests the user additional features to examine in order to differentiate between diagnoses [44,45]. Velikova et al. [46] presents a decision support system that can detect breast cancer based on breast images, the patient’s history and clinical information. To address this goal, they integrate the three approaches to model the knowledge: consistence-based, causal relationships and Bayesian network. MUNIN is a causal probabilistic network for diagnosing muscle and nerve diseases through analysis of bioelectrical signals, with extensions to handle multiple diseases [47,48].

### 2.2. Troubleshooting

Mcilraith [49] presented the theoretical foundation for sequential diagnosis, where a probe is a special case of a *truth test*, which is a test checking if a given grounded fluent is true. This process is similar to clinical evaluation, where the PT performs tests to discriminate between diagnoses. Physiotherapy clinical evaluation is also similar to the active diagnosis problem [50,51], which is the problem of how to place *sensors* in a discrete event system to verify that it is diagnosable, given a set of observations. A very similar problem is the sensor minimization problem [52], where *observers* are placed on particular events to make sure the system is diagnosable and the number of observers is minimized [53]. None of these works reasons about scenarios in which the true state of a component can be masked by other components to return inconsistent values upon probing. Mirsky at el. [54] discuss a similar problem, where the presence of a component in the true hypothesis can be inferred by probes, but they do not reason about a scenario where a specific probe returns one value, while its true state is the opposite value, as discussed in our work.

To reduce the number of hypotheses, McSherry et al. [55] propose a mechanism for independence Bayesian framework. The strategy they propose searches for lower and upper bounds for the probability of the leading hypothesis as the result of each test is obtained. Rather than a myopic minimum entropy strategy they propose efficient techniques for increasing the efficiency of a search for the true upper or lower bound for the probability of a diagnostic hypothesis.

Algorithms for minimizing troubleshooting costs have been proposed in the past. Heckerman et al. [56] proposed the decision theoretic troubleshooting (DTT) algorithm. Probing and testing are well-studied diagnostic actions that are often part of a troubleshooting process. Probes enable the output of internal components to be observed, and tests enable further interaction (e.g., providing additional inputs) with the diagnosed system, providing additional observations (e.g., observing the system outputs). Placing probes and performing tests can be costly, and thus the challenge is where to place probes and which tests to fix the system while minimizing these costs. The intelligent placement of probes and the choice of informative tests have been addressed by many researchers over the years [6,57,58,59,60,61,62] using a range of techniques including greedy heuristics and information gain, which is calculated by comparing the entropy of the hypothesis set before and after a probe is placed [57]. This approach allows for a clear and straightforward mathematical representation of complex systems that can be analyzed to provide completeness, soundness and other computational guarantees. Due to this reason, in this paper we follow the information gain approach and adapting it to handle hidden fault states of the components in the system.

### 2.3. Summary and Our Contribution

In the light of previous work we can see that medical diagnosis is a highly researched area. Most of the previous works can be divided into three approaches: model-based, data-driven and knowledge-based. The main model-based approaches are consistency-based, causal reasoning and Bayesian networks. In many cases the diagnosis method depends on the information available to the researcher. Not always experts exist to help in designing a rule-based system or a model, nor there is enough historical data which can be exploited to generate a classifier or to learn probabilities.

In this work we used expert PTs to generate a model of the the upper human body which is innervated by the nerve roots C−3 to T−1. Unfortunately, we did not have historical data to learn the probabilities of each component to damage nor the conditional probabilities between components. As far as we know, this knowledge is not modeled for neuro-muscular diagnosis in physiotherapy for this part of the body. Therefore, our diagnosis and troubleshooting algorithms assume uniform distribution. Obviously, this can be easily changed given probabilistic knowledge.

The main contribution of this paper is a consistency-based diagnosis and troubleshooting tool, especially for trainee PTs, that includes: (1) An interactive visual model, which helps a PT to see the connections between the nerve roots, nerves, muscles and dermatomes. (2) A diagnosis process which assists the PT to generate hypotheses, given the patient’s symptoms. (3) A troubleshooting process that proposes the PT a sequence of tests to discriminate the hypotheses and focus on the correct one. To the best of our knowledge, this is the first tool that combines these components to assist trainee PTs.

## 3. Architecture and Interface

The system is constructed of several components in a client-server framework, which is designed to allow high usability and applicability for PTs in their clinical evaluations. These components are depicted in Figure 1. A relational database (DB) is implemented using MSSQL to store the connections between the different entities. The server side is ASP.NET and it connects directly to the DB. After a connection is established, an Entity Framework is used to map the tables into objects, to allow easier and faster manipulations on the data. Finally, the client side is implemented using HTML, Javascript and JSON. The system’s home page is web-based, which allows the user to navigate to one of the following modules:

**Maps** The purpose of this module is to provide visualization of the anatomical entities in the human body, while allowing to focus on different structures. This module contains an inner navigation bar, to choose between one of several views: root nerves, nerves, muscles, dermatomes and relations. All maps but the latest focus on different component types and present the names of the relevant components on an illustration. The relations map is a hierarchical representation of the connections between the different entities. It is similar to the relationships graph in the relationships module, but its visualization focuses only on a specific component at a time. An example of this representation is shown in Figure 2. Clicking on one of the nodes constructs a graph of the dependencies of this node.**Relationships** The purpose of this module is to allow a thorough investigation of the relations between the different components of the body. The navigation through the different components can be performed either by using a drop-down list and choosing a specific item from it, or by clicking directly on a node in the graph. The complete relationship graph is presented in Figure 3. This module enables to dynamically navigate from one node to another, a feature which allows the PT to investigate causal connections.**Diagnosis** The purpose of this module is to diagnose the patient, given a list of symptoms. The initial screen of this module is shown in Figure 4. This screen contains two lists of possible symptoms—muscles and dermatomes—which can be added by the PT. When the PT finishes adding initial symptoms, a click on the “Diagnose” button will trigger a recommendation for the next component to check, and then the system requests the PT to update whether the test passed or failed (the component works as expected or not). At any point, the PT can choose to stop this process and receive a list of the remaining diagnoses.

## 4. Technical Description

In this section we will describe technical details about the different parts of *PhysIt*. Specifically, we will describe the model we used (Section 4.1), the diagnosis algorithm (Section 4.2) and the troubleshooting process (Section 4.3).

### 4.1. Model Description

The first feature of *PhysIt* is a model of the entities involved in a physiotherapy diagnosis. We elicited a model of the upper human body which is innervated by the nerve roots C−3 to T−1, or from head to the upper part of the torso. We acquired the information through interviews with senior PTs and data gathering from physiotherapy graduate students. The entities we modeled are *Nerve roots*, *nerves*, *muscles* and *dermatomes*. The relations between the different entities are described in Figure 5:
**Nerves** are the common pathway for messages to be transmitted to peripheral organs. A damaged nerve can cause paralysis, pain or numbness in the innervated organs.**Nerve Roots** are the initial segments of a nerve affected by the central nervous system. They are located between the vertebrae and process all signals from the nerves. A damaged nerve root can cause paralysis, weakened movement, pain or numbness in vast areas of the body.**Muscles** are soft tissues that produce force and movement in the body. A damaged muscle can cause weakness, reduced mobility and pain.**Dermatomes** are sensory areas along the skin, which are traditionally divided according the relevant nerve roots that stimulate them. A damaged dermatome is usually caused by a scar or burn and can cause pain, numbness or lack of sense.

As can be observed from the list of entities, some of the symptoms overlap each other. Tingling sensation at the tip of the index finger can be related either to a problem in a nerve root labeled C−7, to a burn in the relevant dermatome DC−7, or to a problem in a median nerve. Since this work only focuses on damages to the peripheral nervous system or muscular system, we assume that a symptom that is expressed in a dermatome is a signal to a damage in either a nerve root or a nerve. Moreover, the tingling sensation is a cue related to a dermatome, but the dermatome itself is assumed to be healthy. We will elaborate more on this issue later.

The anatomical data for creating this model was elicited by us using physiotherapy students and approved by faculty members with clinical experience. We mapped the relations between all pairs of entities in terms of functionality. A fragment of the elicited relational model is presented in Figure 6. The nodes represent the different components, the colors indicate their type and an edge indicates that one node influences or influenced by the other node associated to it.

When modeling the human body in the context of the physiotherapy diagnosis process, the following comments and constraints should be considered:The observations are symptoms or cues, reported by the patient or by the PT.Each observation is a signal that can be influenced by more than one component in the system. For example, a tingling sensation in the plantar side of the thumb is a signal from a specific dermatome called DC−6, which can be influenced by a problem in the respective root nerve C−6, or from a nerve called radial.The health state of a component cannot be directly evaluated, but must be inferred from observations. Thus, to test the radial nerve described above, the PT will try to cause a tingling sensation in the thumb or to find weakened movement in the hand extensor.The outcome of a test does not always directly implies the health state of a component, but can be masked by other components in the system. For example, inability to perform shoulder extension is a signal related to the deltoid muscle, but even when the deltoid is healthy, the extension might fail due to a problem in the radial nerve or the nerve root C−6.

### 4.2. The Diagnosis Process

We adapt a model-based diagnosis approach to handle the diagnosis process in *PhysIt*. Let us formalize the diagnosis process as a MBD problem [2,3]. Typically, MBD problems arise when the normal behaviour of a system is violated due to faulty components, indicated by certain observations.

**Definition** **1**(MBD Problem). *An MBD problem is specified by the tuple 〈SD,COMPS,OBS〉 where: SD is a system description, COMPS is a set of components, and OBS is the observations. SD takes into account that some components might be abnormal (faulty). This is specified by the unary predicate h(·). h(c) is true when component c is healthy, while ¬h(c) is true when c is faulty. A diagnosis problem arises when the assumption that all components are healthy is inconsistent with the system model and the observation. This is expressed formally as follows*
$$SD\wedge \underset{c\in COMPS}{⋀}h\left(c\right)\wedge OBS⊢⊥$$

Diagnosis algorithms try to find *diagnoses*, which are possible ways to explain the above inconsistency by assuming that some components are faulty.

**Definition** **2**(Diagnosis). *A set of components Δ is a diagnosis if*
$$SD\wedge \underset{c\in \Delta }{⋀}\neg h\left(c\right)\wedge \underset{c\notin \Delta }{⋀}h\left(c\right)\wedge OBS\neg ⊢⊥$$

There may be multiple diagnoses for a given problem. A common way to prioritize diagnoses is to prefer *minimal diagnoses*, where a diagnosis Δ is said to be *minimal* if no proper subset $\Delta^{\prime }\subset \Delta$ is a diagnosis. In this work we will focus on finding minimal diagnoses. Let us formalize the neuro-muscular diagnosis in physiotherapy in terms of a MBD problem.

#### 4.2.1. COMPS

In our model, COMPS is a set of all nerve roots, nerves, muscles and dermatomes. Each c∈COMPS has a health state described by h(c)∈{True,False}. However, since the physiotherapy clinical evaluation only discusses the neuro-muscular systems rather than other pathologies such as skin burns, the dermatomes are assumed to be healthy components that are only used for testing other components. This means that for each dermatome d∈COMPS, it holds that h(d)=True.

#### 4.2.2. OBS

The observations, OBS in our model, are the patient’s weakened motions or defected sensations. Typically, a patient is not connected to sensors that measure the weakened motion or defected sensation. Instead, the PT stimulates the component, for instance a muscle, and observes whether it is defected. To formalize the observation, let us define a test of a component. Given a component *c*, we define the predicate testOK(c)∈{True,False}, where testOK(c)=True indicates that the test successfully passed, meaning, the motion or the sensation are not defected. Consequently, OBS⊆{testOK(c)∣c∈COMPS}.

#### 4.2.3. SD

SD represents the behaviour of the components as well as the influence of each component on the others. Obviously, it is very hard to formalize the behaviour, even for experts. For example, a problem in the radial nerve might cause pain in the shoulder area, but it can also cause numbness, weakened movement or none of these symptoms. Nevertheless, it is possible to formalize that once the inputs of a component are proper and the component is healthy, then we expect to get proper outputs. Let in(c) and out(c) be the input and output of a component, respectively. We define the predicate ok(in(c)), where ok(in(c))=True indicates that the input of component *c* is proper. In the same way we define the predicate ok(out(c)). If a component has more than a single input (output) we will add the index to the input (output), $in_{i}\left(c\right)$ ($out_{i}\left(c\right)$). Also, assume $c_{n}$ and $c_{m}$ represent the number of inputs and outputs of component *c*, respectively. Then the next formula states the behaviour of a component:$$\begin{array}{c} \forall c\in COMPS:\left(\underset{i\in \left\{1,...,c_{n}\right\}}{⋀}ok\left(in_{i}\left(c\right)\right)\wedge h\left(j\right)\right)\to \underset{i\in \left\{1,...,c_{m}\right\}}{⋀}ok\left(out_{i}\left(c\right)\right) \end{array}$$

In addition, we formalize how a proper output influences a test. Intuitively, proper outputs entails that a test passed successfully. Thus we add the following formula:$$\begin{array}{c} \forall c\in COMPS:(\underset{i\in \left\{1,...,c_{m}\right\}}{⋀}ok\left(out_{i}\left(c\right)\right))\to testOK\left(c\right) \end{array}$$

Finally, to formalize the connections between the components, we use the inputs and outputs of the components. If, for instance, the first output of component $c_{i}$ is the first input of $c_{j}$ we add a next equality: $out_{1}\left(c_{i}\right)=in_{1}\left(c_{j}\right)$.

We would like to draw the attention of the reader to two conclusions arising from this model:**Transitivity:** for a given component *c*, if (1) h(c)=True and (2) every component $c^{\prime }$ that affects *c* ($out\left(c^{\prime }\right)=in\left(c\right)$) is healthy ($h(c^{\prime })=True$) and (3) the inputs of $c^{\prime }$ are proper $(ok(in\left(c^{\prime }\right)$), then it must hold that testOK(c)=True.**Weak Fault Model (WFM):** in this model we describe only the healthy behaviour of a component rather than its faulty modes. Thus, we cannot conclude anything about the success of a test (testOK(c)) in case the component is faulty (h(c)=False). In addition, in case a test passed successfully, we cannot conclude that the component checked by this test is healthy. Only in case that a test failed, we can conclude that the tested component or one of its antecedents is faulty.

Once we formalized the problem in terms of an MBD, we can use any off-the-shelf MBD algorithms. MBD algorithms can be roughly classified into two classes of algorithms: conflict-directed and diagnosis-directed [63]. A classical conflict-directed MBD algorithm finds diagnoses in a two-stage process. First, it identifies conflict sets, each of which includes at least one fault. Then, it applies a hitting set algorithm to compute sets of multiple faults that explain the observation [2,4,64]. These methods guarantee sound diagnoses (i.e., they return only valid diagnoses), and some of them are even complete (i.e., all diagnoses are returned). However, they tend to fail for large systems due to infeasible runtime or space requirements [5].

Diagnosis-directed MBD algorithms directly search for diagnoses. This can be done by compiling the system model into some representation that allows fast inference of diagnoses, such as Binary Decision Diagrams [65] or Decomposable Negation Normal Form [66]. The limitation of this approach is that there is no guarantee that the size of the compiled representation will not be exponential in the number of system components. Another approach is SATbD, a compilation-based MBD algorithm that compiles MBD into Boolean satisfiability problem (SAT) [5,67], and then uses state-of-the-art SAT solvers to find the possible diagnoses. We follow a similar line of work here, but instead of a classical SAT solver we use a conflict-directed algorithm, which allows us to find conflicts in polynomial time in our domain by using a Logic-based Truth Maintaining System [68]. The number of conflicts and their size, in our domain, are not so big and enable a standard hitting set algorithm to compute the diagnoses in a reasonable time.

### 4.3. The Troubleshooting Process

While the diagnoses computation is feasible, the diagnosis process may still produce a large set of possible diagnoses. To assist the PT to disambiguate between the diagnoses and focus on the root cause of the pain, the third feature of *PhysIt* enables a troubleshooting process. The challenge in troubleshooting is which test(s) to choose. This process iteratively proposes tests that can discard incorrect diagnoses and focus on the root cause. We adopt the information gain approach to choose the tests to perform [6,57,60,61,62].

Algorithm 1 presents this process. After running the diagnosis algorithm, it creates a list of possible tests (probes) which include all the components in the diagnosis sets (line 1). It then chooses the probe that gives us the highest information gain (line 1). In practice, we broke ties randomly.

After querying about the best probe, the algorithm updates the diagnosis set: if the test successfully passed (probe’s output was true), there is nothing to update (since the model is a weak fault model). Otherwise, it means that either the probed component or one of its affecting components is faulty. Hence, the algorithm removes all the diagnoses that do not contain the tested component or one of its inputs. Lastly, it updates the list of the remaining probes accordingly. This process continues until the diagnosis set *D* is not shrunk by the probes anymore. At the end of the process, the algorithm returns a list of the remaining diagnoses.
**Algorithm 1:** Probing Process
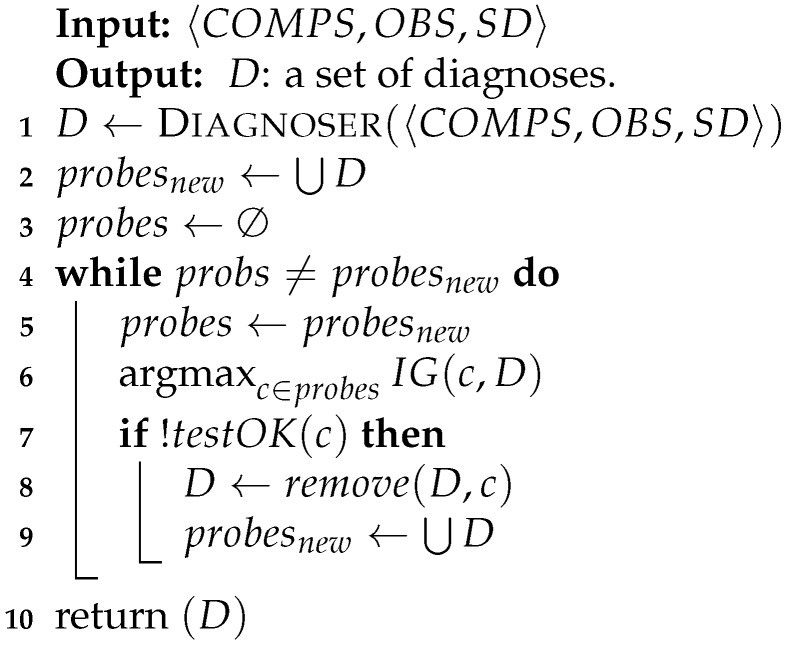


The information gain calculation is a standard metric for quantifying the amount of information gained by testing a component [69]. This can be achieved by comparing between the entropy of the diagnosis set before and after the test. The entropy of the diagnosis set *D* is defined as
(1)$$Ent\left(D\right)=-\sum_{\Delta \in D}P(\Delta )\cdot log(P(\Delta ))$$
where P(Δ) is the probability of the diagnosis Δ. If the components fail independently of each other, then $P\left(\Delta \right)=\prod_{c\in \Delta }P\left(c\right)$, where P(c) is the probability of component *c* to fail. Without prior information, a common assumption is a uniform distribution of the components to fail [10,11]. The information gain from a probe is the difference between the entropy of the set *D* before the test of *c* and the entropy of the set $D^{\prime }$ remains after the test: $IG\left(D|c\right)=Ent\left(D\right)-Ent\left(D^{\prime }\right)$.

## 5. Performance Analysis

We evaluated the diagnosis correctness and the troubleshooting performance in *PhysIt* using empirical analysis of the outputted diagnoses, based on metrics from information retrieval and diagnostics. These metrics were evaluated both on simulated scenarios, and on case studies representing common scenarios we received from PTs. We first present the methodology of the scenario generation (Section 5.1) and the results on these scenarios (Section 5.2). Then we present the results on scenarios based on real-world clinical experience (Section 5.3).

### 5.1. Scenario Simulator

In order to evaluate the system, we built a simulator that checks the system’s accuracy and efficiency using different metrics. The simulator has several steps in the fault injection and observation process. At first, the simulator chooses 1 to 5 faulty components, randomly. These components are used, at the end of the diagnosis process, as a ground truth to check the correctness of the diagnoses outputted by our diagnosis algorithm. We name these injected faulty components as “the real diagnosis”.

Next, the simulator collects all components that can be relevant to the real diagnosis: This set includes all the components that were injected as faulty, and the set of components that can be affected by them. For example, nerve root C−6 is connected directly to Radial, Median and other nerves and connected indirectly to Brachialis, Extensor Carpi Ulnaris and other muscles. In this case, the root nerve C−6 is above all in the hierarchy, meaning that any of the components found below it can be affected by it.

Then, the simulator labels these potentially affected components with a value of !testOK with a probability of 0.5. This labeling simulates the answer of a real TP, if the component will be tested in the troubleshooting process. All other components automatically get the value testOK for their test. The simulator makes sure that every component in the real diagnosis has at least one symptom that explains its presence and sets the value of this symptom to !testOK. This step is designed to make sure the completeness of the diagnosis process and that it will not miss the real diagnosis.

At last, out of the set of the symptoms labeled with !testOK, the simulator chooses symptoms that will form the observation set of the real diagnosis. We set the number of observations to be blocked from above by the cardinality of the number of faulty components. For example, in case of four faulty components, the range of the observation set size is between 1 to 4.

### 5.2. Results

We modeled 75 components in the system. We ran the simulator on all possible faults with a single component, and randomly created additional 150 instances per fault cardinality for cases with 2–6 components. In total, we got 825 instances. Out of these instances, 270 diagnoses contained two of more faulty components with a shared affecting component. We discarded these cases, since they cannot be considered under the assumption of minimal cardinality. Thus, the simulator finally outputted 555 different cases. We analyzed the results with several metrics:

#### 5.2.1. Diagnosis Set Size

This metric measures the outputted set of diagnoses before and after the troubleshooting process. As seen in Figure 7, the number of diagnoses grows exponentially with the number of reported faulty components. Blue and green bars refer to the diagnosis set before and after the troubleshooting process, thus it can be seen that the troubleshooting process succeeds in decreasing the number of diagnoses even by a half. The more faulty components the more effective the troubleshooting algorithm is in reducing the number of diagnoses.

#### 5.2.2. False Positive Rate (FPR)

This metric measures the FPR of the outputted set of diagnoses before and after the troubleshooting process. FPR is measured for each diagnosis separately. The formula of this metric is: FPR=FP/N=FP/(FP+TN), where FP is the number of components in the diagnosis that are not really faulty and TN is the number of components that are not in the diagnosis and are healthy. To compute the FPR of the whole set of diagnoses, we computed the weighted FPR, by multiplying the FPR of each diagnosis by its probability. Since the probabilities of the diagnoses are normalized the computation of the weighted FPR is correct.

The *x*-axis in Figure 8 refers to the number of faulty components while the *y*-axis refers to the FPR value. Blue and green bars refer to the diagnosis set before and after the troubleshooting process, correspondingly. The lower FPR the better. There is a positive correlation between the number of faulty components and the FPR value, since the more faulty components the more diagnoses contain false positive components. Nevertheless, we can see two positive results: (1) the FPR is low even when the faulty components number increases, (2) the troubleshooting process reduces the FPR.

#### 5.2.3. Area Under the Curve (AUC)

To explain this metric we should define first the term **Health State**, which has recently proposed by Stern et al. [10,11]. The health state indicates the probability of each component to be faulty, given a set of diagnoses *D* and a probability function over them *p*:(2)$$H\left(c\right)=\sum_{\Delta \in D}p\left(\Delta \right)\cdot {𝟙}_{c\in \Delta }$$
where ${𝟙}_{c\in \Delta }$ is the indicator function defined as:$${𝟙}_{c\in \Delta }=\left\{\begin{array}{cc} 1 & c\in \Delta \\ 0 & \mathrm{otherwise} \end{array}\right.$$

Based on the health state, Stern et al. propose the AUC metric. The AUC is usually used in classification analysis to determine if the model predicts the classes well. In order to calculate the AUC value, we calculate the FPR and TPR of 11 thresholds values, 0 to 1 in hops of 0.1. Each threshold value creates a pair of values (FPR and TPR) which eventually becomes a point on the Receiver Operating Characteristic curve (ROC). The AUC is the area under the ROC curve. The higher the AUC the more accurate health state. Each threshold determines the set of components for which the FPR and TPR are calculated. All components have a higher health state than the threshold are taken into consideration.

As seen in Figure 9, the *x*-axis refers to the number of the faulty components while *y*-axis refers to AUC value. Blue and green bars refer to the diagnosis set before and after the troubleshooting process, correspondingly. There is a negative correlation between the number of faulty components and the AUC, since the number of diagnoses grows with the number of faulty components and thus the health state is less accurate. Furthermore, the AUC of the health state computed for the set of diagnoses before the troubleshooting process is lower than the AUC calculated after the troubleshooting process. This shows the benefit of the troubleshooting process.

#### 5.2.4. Top-K

This metric is known in the information retrieval literature. It checks whether the real diagnosis exists in the top-*K* diagnoses returned by the algorithm, where *K* is a number between 1 to 5. The diagnoses are ranked in a decreasing order of their probability. As seen in Figure 10, the *x*-axis refers to the *K* value while the *y*-axis refers to the ratio of instances that had the faulty components in the top-*K* diagnoses. Blue bars refer to initial diagnosis, while final diagnosis are presented by green bars. As the value of *K* increases, the chance to be in the top *K* increases too. It is clear that the final set of diagnoses shows better results than the initial set which means that the troubleshooting algorithm is indeed a helpful tool to reduce the size of the diagnosis set while improving the localization of the real diagnosis.

All of the above experiments were conducted under the strict assumption that a faulty component may be assigned !testOK with a probability of 0.5. In practice, this probability is expected to be closer to 1 than to 0.5. Therefore, all experiments were repeated such that the simulator always assigns !testOK to a faulty components and the components it affects. Table 1 summarizes the results of the evaluated metrics so far, using this relaxed assumption, in order to show the real potential improvement of using this system. The rows represent the metrics and the columns represent the number of faulty components. For each metric and cardinality, we compared the initial and final values and present the improvement in the metric in percentage. This table emphasizes that the bigger the cardinality, the more difficult the problem is to solve. However, the benefit of using the troubleshooting process is clear: the process manages to remove irrelevant diagnoses (according to the improvement in the wasted cost and top-5 metrics), without hindering the correctness of the results (since the FPR only improves). Moreover, the improvement of the troubleshooting becomes greater as the number of faulty components increases.

#### 5.2.5. Comparing to Random

Finally, we show the benefit of the troubleshooting algorithm comparing to a random approach. The random approach chooses randomly the next component to test from a set which includes the union of all the diagnoses. Obviously, both the information gain algorithm as well as the random algorithm will finally invoke the same set of tests and the final set of diagnoses will be the same. However, the order of invoking the tests is different between the two algorithms, and might affect how fast the diagnosis set is reduced. Figure 11 shows the influence of the order of the tests (*x*-axis represents the number of tests) on the number of diagnoses. As shown, the troubleshooting algorithm which uses the information gain reduces the size of the diagnosis set faster than random. Even after using a single probe, the random algorithm reduces the number of diagnoses by 38%, and the information gain algorithm manages to reduce it by 47%. This is a significant difference across the examined cases (*p* < 0.01). We repeated this experiment for different cardinalities (number of faulty components), and the reduction trends remain the same for all cardinalities (1 to 5).

### 5.3. Real-World Scenarios

With the help of experts from the Physiotherapy Department in Ben-Gurion University of the Negev, we modeled 17 representative scenarios of common cases, which are in use in physiotherapy anatomy exams. As these are written scenarios and not clinical evaluation performed on real patients, the value of some of the components is unknown, and the results of any test performed in order to reduce the possible diagnosis set will have to be simulated. Simulating test results for this lack of values will not benefit new insights beyond the ones already received from the simulated cases. Instead, we focus this evaluation on the correctness of the outputted diagnosis set before the troubleshooting process.

In 16 out of the 17 cases investigated, the outputted diagnosis set contained the real diagnosis as reported by the PTs. In a single case, the real diagnosis was not a minimal one - but a combination of two nerve roots C−5 and C−6. According to the constructed model, all the symptoms could be explained exclusively by C−6, so the diagnosis error is redundant. Since our diagnosis algorithm searches for minimal subset diagnoses it missed this diagnosis.

Due to the completeness property of our troubleshooting process, in 16 out of the 17 cases the system managed to decrease the size of the diagnosis set without removing the correct diagnosis. These results show that even in realistic scenarios conducted by experts *PhysIt* found sound diagnoses and succeeded to reduce the diagnosis set without missing the real diagnosis.

## 6. User Study

The promising results of the diagnosis system both on simulated and real scenarios, encouraged us to test the system in a human study, in order to show its ability to assist students in their physiotherapy studies. There is a variety of books and atlases that teach students anatomy [70,71,72]. However, to the best of our knowledge, no system is in use to assist physiotherapy students in the beginning of their clinical studies. For this reason, we devised a user study to evaluate the usefulness of *PhysIt* specifically for students in an advanced stage of their physiotherapy studies.

### Experimental Setup

The experiment consists of simulations of clinical diagnoses with and without the various modules of the *PhysIt* system (maps, relationships and diagnosis), following by a questionnaire to evaluate the students’ experience with the system. We constructed a wrapper to our system with a landing page that can direct the user to the three different modules of *PhysIt* and to a simulator that imitates the diagnosis process.

The simulator begins with a list of symptoms that represent the patient’s complaints at the beginning of a diagnosis process. Then, the participant (the experimenter) could choose a test from a list of dermatomes, muscles, nerves and nerve roots. The simulator simulates the test of the selected component by the physiotherapist and returns whether the test passed successfully (the selected component is healthy) or unsuccessfully. This process is done as long as the experimenter wishes to perform tests. The cases that were chosen for the simulator are based on the 17 expert case studies. As these cases do not elaborate the results of all possible tests, the results of unknown tests were chosen as follows. For a component that is clearly unrelated to the patient’s symptoms, the relevant test returns that the component is healthy; for a component that is clearly related to the patient’s symptoms, the test returns that the component is not healthy; and for a component that might be connected to one of the symptoms, the test result will be chosen at random. The simulated scenario ends when the participant decides on a diagnosis. The participants were not informed with the correctness of their responses, so it will not affect their answers about their experience with the system. A screenshot of the simulator is presented in Figure 12.

The three modules of *PhysIt* that were evaluated are: maps, relationships and diagnosis (see Section 3 for details). The participants were divided into three groups, such that each one of them had an access to a different subset of the system modules. The first group could only use the maps module; the second could use the maps and the relationships modules; and the third could use all of the three modules.

In addition to the simulations and recorded test sequences and diagnoses, the participants were requested to answer a questionnaire about their experience with the system. The questionnaire consists of the following questions:
1.Improve:Did the system improve your choice of tests to perform?

(yes/no)2.Clear:Was the system easy to understand?

(5-point scale)3.Use:Was the system easy to use?

(5-point scale)4.Preference:Which of the components did you use the most?

(choice between available components)5.Open:In your opinion, was there something that was missing in the system?

(open question)

Thirty one participants in the third year of their physiotherapy studies were divided into three groups: The first group consisted of 10 student and received access to the maps module of the *PhysIt* system (the Maps group); the second consisted of 10 students and received access to both the maps and the relationships module (the Relationships group); and the third group consisted of 11 students and received access to all components of the *PhysIt* system (the Diagnosis group).

Figure 13 shows the results of the first question (Improve) and the fourth question (Preference). As seen on the left side of the figure, the Relationships and the Diagnosis modules are considered by the subjects to improve their diagnosis process significantly more than the Maps module (p=0.027 and p=0.012 respectively). The Fleiss’ Kappa agreement between the subjects is 81% in the Relationships group and 66.3% in the Diagnosis group. As seen on the right side of the figure, out of the participants in the Diagnosis group, 55% preferred the diagnosis module over the other modules of the system. Out of the students in the Relationships group, all students preferred the Relationships module over the Maps module. The results of the other general questions (Clear and **Use**) seem to be a slight preference to the diagnosis module over the other modules but they this preference is statistically insignificant. We have also calculated precision and recall for the diagnoses returned by the students compared to the root problem, but these results were insignificant as well.

For the Open question about what is missing in the system, the most common answer was that the system is missing a preliminary layer where patients can describe their symptoms (e.g., *“The patient will complain on a tingling sensation, numbness, pain or weakness, not on a NOT-OK deltoid"*). The patient’s complaints from this preliminary layer might later be connected to other components. Another reoccurring answer complements that the system lacks more detailed diagnoses (*“e.g., the root cause of a problem is Tennis elbow rather than a NOT-OK Extensor Carpi Radialis Brevis"* and *“It would be nice to add to the diagnosis whether this is a chronic or acute condition"*). Overall, it seems like the participants felt that the system over-simplified the diagnosis process, but was still considered useful as an educational tool.

## 7. Conclusions and Future Work

In this work, we presented *PhysIt*, a tool for diagnosis and troubleshooting for physiotherapists. We managed to apply an MBD approach in the real world, using a physiotherapy-related domain. We applied a classical MBD algorithm to compute diagnoses given some symptoms and showed that a troubleshooting process can significantly decrease the number of candidate diagnoses, without discarding the correct diagnosis. Experiments on synthetic scenarios show the benefit of the troubleshooting algorithm. Additional experiments on real scenarios show the potential benefit of *PhysIt* to reduce the set of diagnoses without hindering completeness. A user study conducted with students shows that the system could potentially be in use for physiotherapy studies in the beginnig of clinical training.

From discussing this work with many PTs who are familiar with clinical evaluation and diagnosis, it seems that several desired properties are necessary in the future:A malfunction in the muscle is usually reported by the patient as a mobility issue. Identifying the relevant muscle based on motion disability or pain is part of the clinical evaluation, which is not presented in our model. We intend to extend the system to include “movement” entities and their relations to muscles and nerves.In practice, most tests do not output a binary result and a component can have more states rather than testOK and !testOK. We wish to augment probabilities in our model - both to represent a degree of “faultiness” and to be able to evaluate the impact of batches of tests.As shown in previous papers, abduction with a model of abnormal behaviour is a much better way to deal with medical diagnosis. To this aim we plan to achieve more information about the abnormal behaviour of components and integrate it in our model in order to discard redundant diagnoses.

## Figures and Tables

**Figure 1 diagnostics-10-00072-f001:**
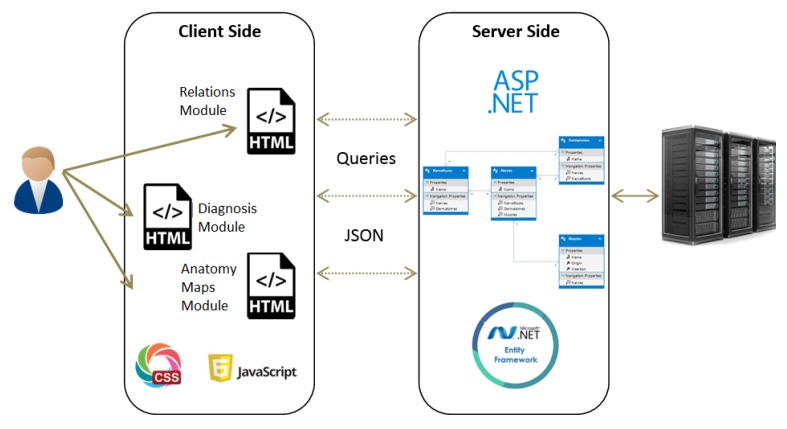
Framework description of the system.

**Figure 2 diagnostics-10-00072-f002:**
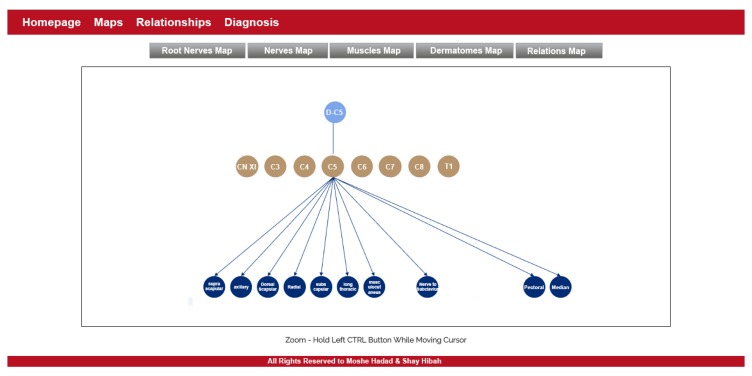
The maps module.

**Figure 3 diagnostics-10-00072-f003:**
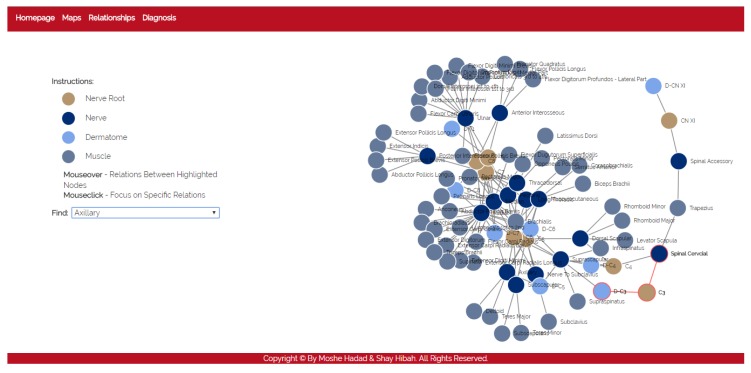
The relationships module.

**Figure 4 diagnostics-10-00072-f004:**
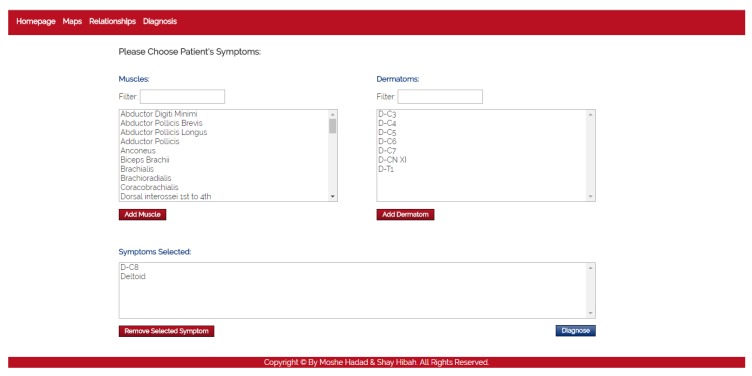
The diagnosis module.

**Figure 5 diagnostics-10-00072-f005:**
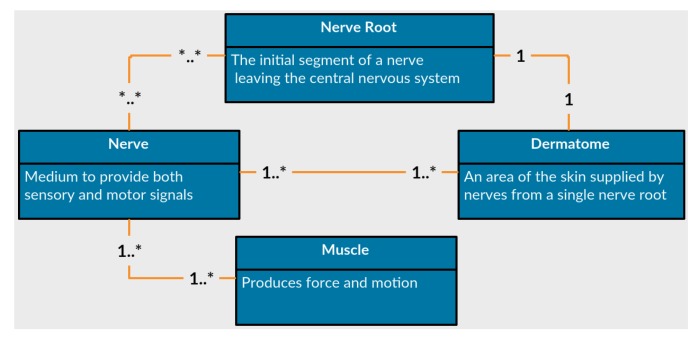
Anatomical entities represented in the diagnosis models.

**Figure 6 diagnostics-10-00072-f006:**
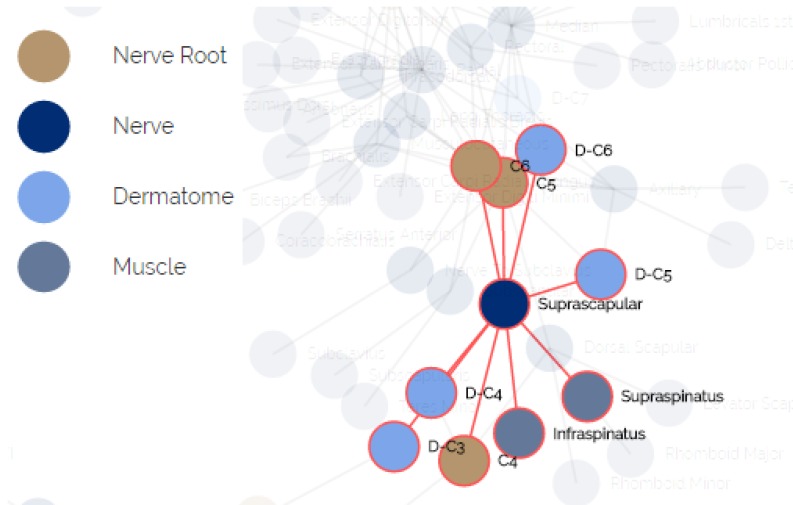
The relational underlying model of anatomical entities.

**Figure 7 diagnostics-10-00072-f007:**
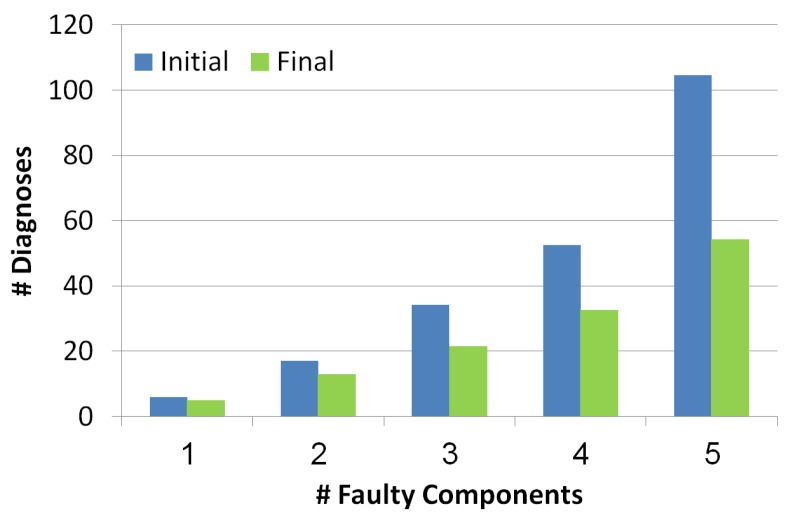
Number of diagnoses before and after the troubleshooting process.

**Figure 8 diagnostics-10-00072-f008:**
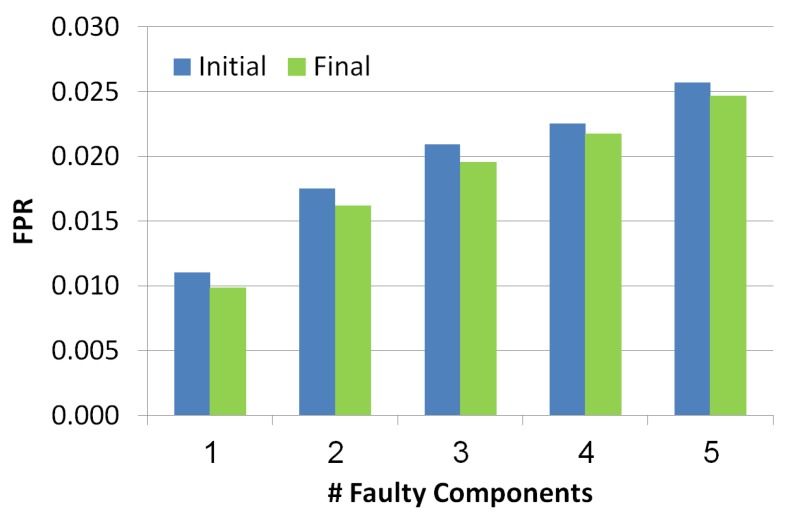
False positive rate of the simulated scenarios.

**Figure 9 diagnostics-10-00072-f009:**
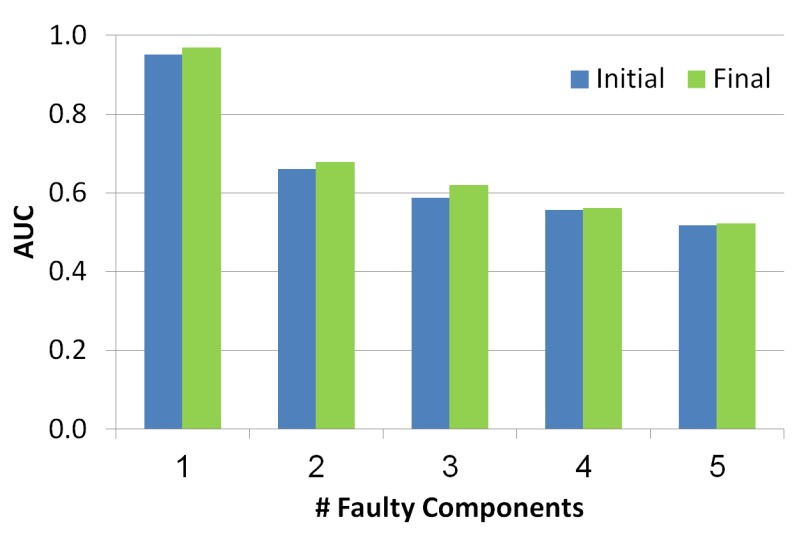
Area under the curve of the simulated scenarios.

**Figure 10 diagnostics-10-00072-f010:**
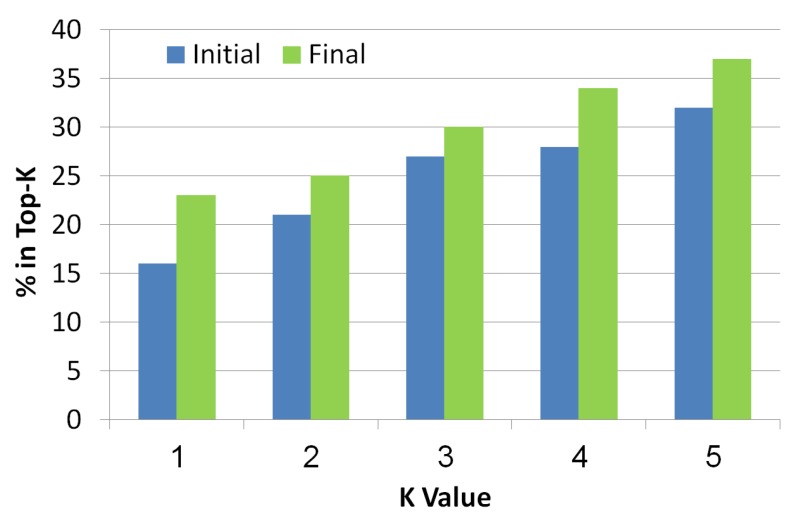
Top-K of the simulated scenarios.

**Figure 11 diagnostics-10-00072-f011:**
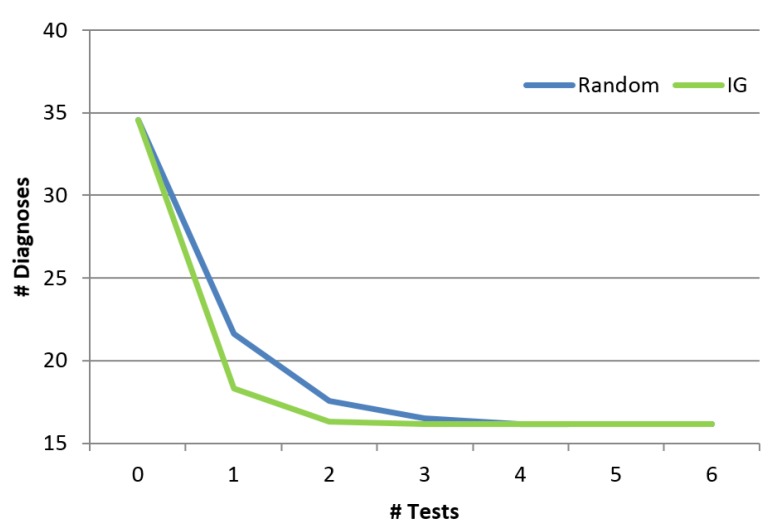
Reduction of the diagnosis set.

**Figure 12 diagnostics-10-00072-f012:**
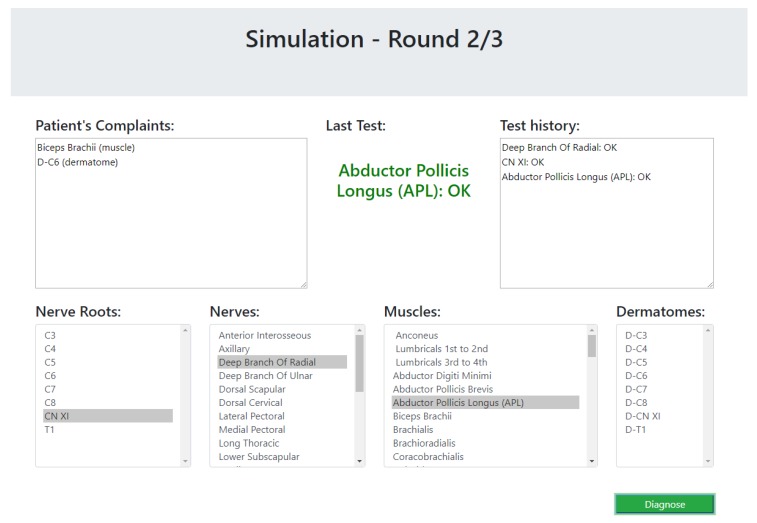
A snapshot of the user study simulator.

**Figure 13 diagnostics-10-00072-f013:**
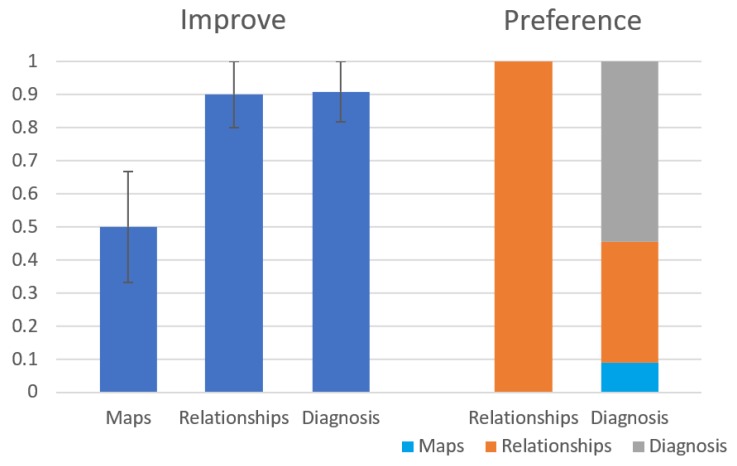
Results for Improve and Preference questions from the user study.

**Table 1 diagnostics-10-00072-t001:** Improvements in metrics per number of faulty components. *—initial value was 0. **—initial and final values were both 0.

Metric	1	2	3	4	5
FPR	0.11	0.08	0.06	0.03	0.04
AUC	0.01	0.05	0.05	0.01	0.03
Wasted Effort	0.15	0.25	0.42	0.44	0.54
Top-5	0.05	0.24	0.67	$1.00{\;}^{*}$	$0.00{\;}^{**}$

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
