# Peer review of "“PhysIt”—A Diagnosis and Troubleshooting Tool for Physiotherapists in Training"

_diagnostics, 2020, doi:10.3390/diagnostics10020072_

Round 1

Reviewer 1 Report

Comments for authors:
The paper presents a model based diagnostic system (PhysIt) to assist the trainee physiotherapists in diagnosis. The results show that the system can significantly reduce the number of candidate diagnosis without discarding the correct diagnosis , and the user study shows that the PhysIt system is helpful in the diagnosis process. The article is clear in structure, rich in content s , an d has potential for application . However, to further improve
the quality, I have some questions and suggestions as follows:
(1) In section 2 (page 5, line 187 ), the author s mention that the intelligent placement of probes and the choice of informative tests have been addressed by many researchers over the years using a range of techniques including greedy heuristics and information gain .Please explain why the information gain approach is used in this article
(2) In section 4 (page 11 , line 333 339), t he author s me ntion three kinds of diagnosis directed MBD algorithms based on compilation, and explains the limitations of the first two methods. Please give a further explanation to the existing problems of the SATbD algorithm.
(3) In section 5 (page 14, line 436), t he author s state that the AUC of the health state computed for the set of diagnoses before the troubleshooting process is higher than the AUC calculated after the troubleshooting process , which seems to contradict Figure 9. Please check it
(4) In section 6 , the author divides the participants into three groups according to the three modules of the system, but there are only about 10 people in each group. Is the number too small?

Author Response

We thank you for the clear and helpful review, and for explicitly stating the main contributions of this work. We changed and updated the manuscript according to your comments, and we refer to each of them below:

Information gain for system diagnosis is defined as using the entropy of the hypothesis set before and after a probe is placed (deKleer and Williams, 1995). This approach allows for a clear and straightforward mathematical representation of complex systems that can be analyzed to provide completeness, soundness and other computational guarantees.  We have included this description in the manuscript as well. We think that this might be a confusing point: our work is similar to SATbD in that it formalized the system as a Boolean satisfiability problem, but instead of using a SAT solver like SATbD, we use  a conflict-directed algorithm, similar to the approach that was used by Rutenburg (1994). We clarified this in the text. Thank you for catching this, there was a confusion in the text, it should state “lower” rather than “higher”. We fixed it in the revised version. We agree that the number of participants in our user study is relatively small. It is a challenging recruitment process, as it required the subjects to be in a specific year and right in the beginning of their clinical studies, which means that recruiting more students would require us to wait another year. We tried to overcome this small number of participants, by introducing the system incrementally to portions of the participants who were assigned to the second (relationships) and third (diagnosis) groups, such that these groups will be able to provide feedback on more than just one component of the system.

Reviewer 2 Report

The aim of the paper should be clearly indicated in the Abstract and the Introduction.

In the abstract authors write: ” This work proposes to assist students in this challenge by presenting three main contributions: (1) A compilation of the neuromuscular system as components of a system in a Model-Based Diagnosis problem; (2) The PhysIt is an AI-based tool that enables an interactive visualization and diagnosis to assist trainee physiotherapists; and (3) An empirical evaluation that comprehends performance analysis and a user study.

In the Introduction, instead, it is stated: “This paper presents a decision support system – PhysIt – which aims to assist the trainee PT in the 30 diagnosis and the troubleshooting processes.

It is confusing, the aim should be unified.

The research methodology should be described in introduction (research steps and research methods).

The quality of figure 2 is insufficient. It should be improved.

Author Response

We thank you for your review. We modified the manuscript according to your comments:

We unified the description in the abstract and introduction such that it is consistent. We added a paragraph about the research methodology to the introduction. We have updated Figure 2.